# The Relationship between the Immigrant Rate and Health Status in the General Population in France

**DOI:** 10.3390/jpm11070627

**Published:** 2021-06-30

**Authors:** Jeanne Perrot, Jean-François Hamel, Antoine Lamer, Mathieu Levaillant

**Affiliations:** 1Methodology and Biostatistics Department, Angers University Hospital, University of Angers, F-49933 Angers, France; jeanne.perrot@chu-angers.fr (J.P.); jeanfrancois.hamel@chu-angers.fr (J.-F.H.); 2UMR_S1085, University of Angers, CHU Angers, University of Rennes, Inserm, EHESP, Irset (Institut de Recherche en Santé, Environnement et Travail), F-49000 Angers, France; 3University Lille, CHU Lille, ULR 2694—METRICS: Évaluation des Technologies de Santé et des Pratiques Médicales, F-59000 Lille, France; antoine.lamer@univ-lille.fr

**Keywords:** health inequality, poverty, health determinants, immigrant

## Abstract

Mostly studied at the individual level, the analysis of immigrants’ health status at a populational level may provide a different perspective to investigate, including social determinants as part of the explanation of the relationship between them and health status in France. We analyzed freely accessible databases curated by French public bodies. The dependent variables were death rate and mean age at death. Immigrant rate and covariates associated with either of the outcomes were explored in univariate and multivariate models. Linear models were used to explain the mean age at death, whereas tobit models were used to explain the death rate. The immigrant rate varied markedly from one department to another, as did healthcare accessibility, population’s age profile, and economic covariates. Considering univariate models, almost all the studied covariates were significantly associated with comes. The immigrant rate was associated with a lower death rate and a lower age at death. In multivariate models, the immigrant rate was no longer associated with age at death but was still negatively associated with the death rate. In France, the departments with a higher proportion of immigrants were those with a lower death rate, possibly because immigrants are attracted to economically thriving areas.

## 1. Introduction

France’s immigrant population has almost doubled over the last 70 years with motivation evolving gradually from employment to geopolitical issues, partially due to the economic slowdown [1,2,3]. As their rate grew and motivations evolved, stakes changed, along with an increase in their unemployment rate. Both of these mutations contributed to deterioration of their health status and consider immigrants’ health as a major social and political issue [4,5,6,7,8]. Studied in many countries, including France [4,9,10,11,12,13,14,15], immigrants’ health is often heterogeneous, but it is almost always worse than the health status of non-immigrants [5,11].

Immigrants can declare less chronic diseases and activity limitations, even after adjustment on socio-economic and demographic characteristics [3]. European studies are congruent and conclude in a worse self-perceived health in immigrant population than in non-immigrant one Sweden and Switzerland. The rate was highlighted as being associated with cardiovascular diseases. Mortality rates appeared almost always higher than the native populations in France, Scotland, Denmark, England, and Wales [5,16]. Among the immigrant population, health status could vary according to country of origin [4]. Immigrants tend to recourse less frequently to general practitioners and specialist physicians [11,17]. This is also suggested to have a more important effect on the immigrants’ health and access to healthcare than on the native population [4,11,17,18,19]. A reduced access to supplementary health insurances and a reduced social integration (including a higher unemployment rate and harder work conditions), along with an underprivileged socio-economic situation, are considered as the main factors explaining the disparities in the health status and healthcare access [3].

The health difference between the immigrant and the native population can finally be explained by immigrants’ individual stories, with a long-term effect of the origin country’s political, economic, and health conditions, as well as the impact of the host country’s socioeconomic conditions [3,4]. The life habits inherited from the origin country but also the new ones acquired in the host country [11], as well as the length of stay in the host country, may contribute to the immigrants’ health status [11,20].

Differences in the immigrants’ health status can be accentuated by an uneven access to healthcare facilities across a given geographical area [21]. Indeed, the use of healthcare services depends not only on health conditions but also on the distribution of healthcare facilities. Mismatch between the distribution of immigrants and that of healthcare facilities may worsen care provision for this vulnerable population.

The objective of the present study was to evaluate the county-level relationship between the proportion of immigrants and health status, while taking account of several socio-economic and health determinants as possible confounding variables. To this end, we analyzed freely accessible data provided by French national public bodies. The death rate and the mean age at death were analyzed as proxies for health status.

## 2. Methods

### 2.1. Data Collection

We analyzed sociodemographic and economic data for 96 French “departments”—an approximately county-level geographic scale (i.e., between cities and regions). The five French overseas departments were not included in the analysis because they differ markedly from mainland departments with regard to the population and legislation.

Data were obtained from freely accessible databases curated by (i) the French National Institute for Statistical and Economic Studies (INSEE [22,23], corresponding to the French population census, indicating exhaustively and at the municipality scale the socio-demographic characteristics of the French population, e.g., number of inhabitants, age, sex ratio, economic status, percentage of population being born abroad with a nationality other than French), (ii) the French Institute for Research and Information in Health Economics (IRDES [21], characterizing, at the life-territory level—a supra-municipal scale—, the type of access to healthcare for the population), and (iii) the French National Family Allowances Office (CNAF [24], reporting exhaustively, and, at the municipality scale, the part of the population receiving income support).

### 2.2. Covariates and Outcome Variables

For each department, we considered the death rate and the mean age at death for the dependent variables. As independent variables, we considered the number of inhabitants [22], the mean age [22], the proportion of males, the proportion of the population receiving income support [24], the poverty rate (the proportion of households with an income below 60% of the median national income [23]), the share of households subject to income tax [23], the Gini index of economic inequality, and the immigrant rate (defined here as the number of immigrants as a proportion of the total population). Immigrant status was defined as being born abroad with a nationality other than French [1].

In order to assess the distribution of health facilities and access to care in each department, we used the IRDES “health territory” (HT) classification published in 2019 by Chevillard et al. [21]: HT1: peri-urban areas with a poor level of access to healthcare; HT2: rural and unattractive areas with vulnerable populations; HT3: retirement and tourism areas that are well endowed with healthcare services; HT4: disadvantaged urban or rural areas with dedicated governmental socio-economic and health programs; HT5: socio-economically heterogeneous city centers that are well endowed with healthcare services; HT6: wealthy cities and peri-urban areas. Given that Chevillard et al.’s classification was built on the municipality scale, we aggregated the data and considered all the inhabitants in each department living in each of the 6 types of area.

The study outcomes were immigrants’ death rate (the number of deaths as a percentage of the inhabitants [25]) and their mean age at death.

### 2.3. Statistical Analyses

The different databases were merged and collapsed at the “department” level by weighting each information collected at the municipal level by the number of inhabitants of this city.

Quantitative variables were quoted as the mean (standard deviation) and were compared using *t*-tests. Factors associated with the death rate and the mean age at death were explored in univariate and multivariate models. Linear models were used to explain the mean age at death, whereas tobit models were used to explain the death rate (as a bounded continuous variable that cannot be lower than 0% or higher than 100%). No variable selection process was used when performing multivariate models to avoid any overfitting issue. All the considered independent covariables were included in these models.

The models’ validity was assessed by analyzing the distribution of the residuals. All tests were two-sided, and the threshold for statistical significance was set to *p* < 0.05. All analyses were performed using Stata software (release 14, StataCorp LLC, College Station, TX, USA).

### 2.4. Patient and Public Involvement

Patients or members of the public had no involvement in the design, or conduct, or reporting, or dissemination plans of the research.

## 3. Results

### 3.1. Description of the Population

The characteristics of the departments are summarized in Table 1. The mean number of inhabitants per department was 700,000 (range: 76,600 to 2,604,000), and the proportion of males was fairly uniform (range: 47.0% to 49.8%). The inhabitants’ mean age varied from one department to another (range: 35 to 48), as did the proportion of immigrants (range: 2% to 30%) and the proportion of inhabitants receiving income support (range: 1% to 10%). The poverty rate ranged from 9% to 28%, the share of households liable for income tax ranged from 40% to 70%, and the Gini index ranged from 0.25 to 0.50.

The prevalence of the various types of health territory varied markedly from one department to another (Figure 1). Some departments are 90% rural, others correspond predominately to retirement and tourism areas, and yet others correspond to socio-economically heterogeneous cities that are well endowed with healthcare services.

In the western departments, for example, the most prevalent type of health territory is the city center (which is well endowed with healthcare services), and between 20–40% of the inhabitants live in this type of health territory.

### 3.2. Univariate Analyses

When considering univariate analyses, the number of inhabitants, the population’s mean age, the percentage of households liable for income tax, the Gini index, and specific health areas were associated with the death rate and the mean age at death. The percentage of family allowance beneficiaries was only associated with mean age at death. A higher immigrant rate was significantly associated with both a lower death rate and lower mean age at death.

### 3.3. Multivariate Analysis

The covariates considered in our analyses were highly correlated (Figure 2), which suggests that many were confounding variables. The death rate was significantly and positively associated with the mean age of the population, the proportion of inhabitants receiving income support, the proportion of rural health areas, and the proportion of socio-economically heterogeneous urban areas. The immigrant rate was also significantly and negatively associated with the death rate (Table 2)—albeit with a two-fold reduction in the effect size, relative to the univariate analyses (Figure 3).

In a multivariate analysis, the mean age at death was significantly associated with the mean age of the population, the proportion of inhabitants receiving income support, and the Gini index but not the immigrant rate (Table 3).

The correlation between the studied variables is represented graphically in Figure 2.

Income support: proportion of the population receiving income support. Income tax: proportion of households liable for income tax. For the definition of the health areas HT1 to HT6, please refer to Section 2.

## 4. Discussion

In the present study of the relationship between the immigrant rate and health status in France, we found that several variables differed markedly from one department to another: the immigrant rate, the proportion of income support beneficiaries, the poverty rate, the proportion of households liable for income tax, and the Gini index. These observations were in line with previous studies and showed that the immigrant rate was higher in large urban areas [1]. The degree of inequality was associated with a higher death rate and a lower mean age at death. Likewise, a high proportion of inhabitants receiving income support was associated with a low mean age at death. These relationships have been described previously and are independent of the mean income level [26]. Similarly, income inequality is associated with the mean life expectancy [27,28]. It has been reported that differences in income and socioeconomic status respectively account for 42.5% and 16% of the differences in self-assessed health between immigrants and natives [3].

The relationship between the immigrant rate and the death rate was unexpected: even when adjusting for confounders, a higher proportion of immigrants in a given department was associated with a lower death rate. This phenomenon can be explained either by a heterogeneous distribution of immigrants across France or better health status among immigrants than among natives, i.e., a “healthy immigrant effect” or “immigrant paradox” [15,20,29]. The latter is supposedly due to selective immigration [13], better life habits [9], or a “salmon bias” in which older or sick immigrants return to their country of origin and, thus, leave younger, healthier immigrants in the host country [7,12]. In fact, several studies have found that health status is worse for immigrants than for natives. In some cases, this difference is no longer statistically significant after adjustment for socioeconomic conditions [3]. Nevertheless, comparisons of literature data are complicated by interstudy differences in population and territorial characteristics. In order to provide a wise comparison to this study results, the same method may be applied in different countries to be able to identify a French effect more than an immigrant effect. Furthermore, inferences about individual characteristics should not be deduced from inferences about the group to which those individuals belong [30]. Conversely, the risk of creating an ecological fallacy is high if territory heterogeneity is not taken into account [31]. Several territory covariates (such as income and access to health facilities) were considered in the present study. As the French areas with the highest immigrant rates are those with lowest death rates, French policies should be adapted: specific policies may be needed toward immigrant population in territory that are not the one usually targeted by common programs.

As mentioned above, Chevillard et al. characterized French areas with regard to their access to primary healthcare facilities. This classification is based on 32 selected indicators and facilitates the assessment of measures designed to attract and retain general practitioners in underserved areas. The inclusion of this classification in our multivariate analysis enabled us to mitigate the well-known effect of lower recourse to care [32], which is known to be due (at least in part) to a specific interaction between healthcare workers and immigrants [33]. Even after adjusting for access to primary healthcare facilities in the departments, we still found an association between the immigrant rate and the death rate.

The health areas studied here are heterogeneous; several departments are almost exclusively composed of rural and unattractive areas with vulnerable populations, others contain large retirement areas, and yet others encompass wealthy urban and peri-urban areas that are well endowed with healthcare services. This illustrates the uneven access to healthcare in France and suggests that the areas with the lowest level of access are not those with the highest immigrant rate.

The present study has several limitations. First, our definition of immigration was that used by INSEE (one of the database curators). All people born abroad as a foreigner were therefore considered to be immigrants—even if they had been granted French citizenship since their arrival. Secondly, we did not have information on the immigrants’ length of stay in France. This may have introduced bias because according to Pérez et al., immigrants tend to acquire the same health status as natives after several years in the host country [34,35]. Managing to include the immigrant time of stay on the host country may help to understand the relationship found. It may also be a useful indicator to evaluate the policies implemented by France to integrate immigrant population either in their health system but also along all the sociodemographic conditions explored. Thirdly, the French legislation prevented us from studying the impact of the immigrants’ ethnicity, which can be associated with variations in health status. Fourthly, we did not investigate socioprofessional categories, even though the Whitehall studies performed by Marmot et al. highlighted a social gradient [3,8,27]: the all-cause death rate was lower in people occupying the highest socioprofessional positions. However, the poverty rate could be considered as a proxy for the social rank in our adjusted model, which would bring our results into line with Marmot et al.’s findings. Lastly, several covariates were highly correlated with each one, this might had created collinearity. However, all of our models’ r-square coefficients were greater than 0.8—meaning that most of the variability in the endpoint was explained by the covariates considered and suggesting all the major covariates explaining the death rate or the age at death were included in our analyses.

## 5. Conclusions

Even though the relationship between immigration and health status is well known at the individual level, it warrants further investigation at the population level. The territorial distribution of immigrants is closely related to socio-economic and health characteristics. Despite the precarious health conditions of many immigrants, the French areas with the highest immigrant rates are those with lowest death rates. Such results may be useful to adapt health programs toward immigrant population across French territory.

## Figures and Tables

**Figure 1 jpm-11-00627-f001:**
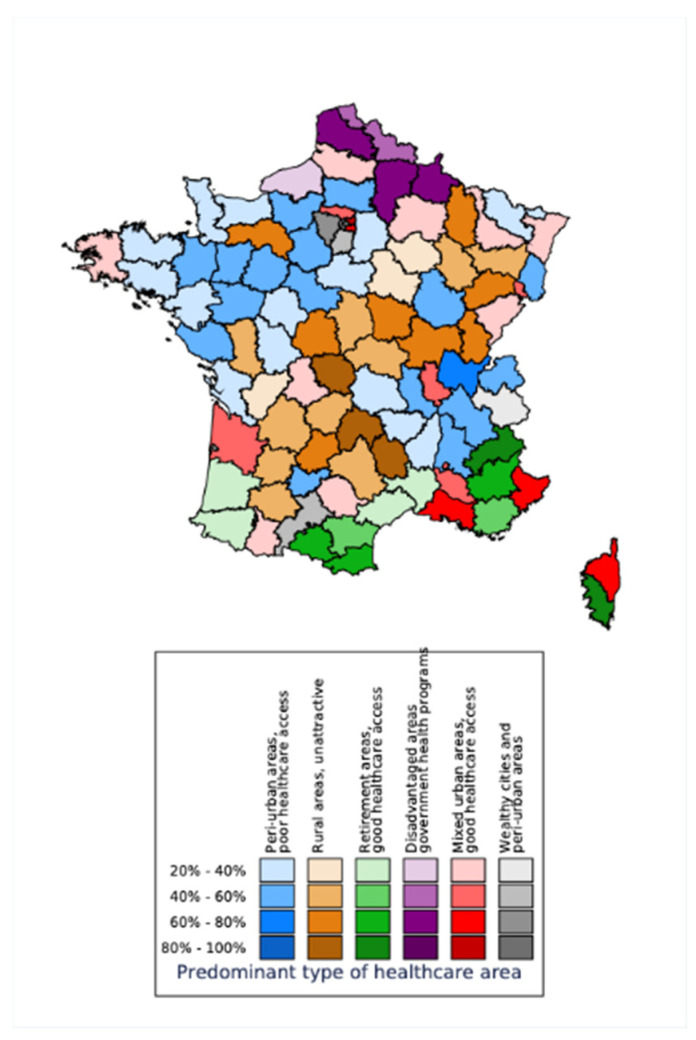
Access to healthcare facilities by department, according to Chevillard et al.’s classification. The color of each department is based on the most prevalent type of health territory. The color intensity (light/dark) corresponds to the proportion of inhabitants living in the most prevalent type of health territory.

**Figure 2 jpm-11-00627-f002:**
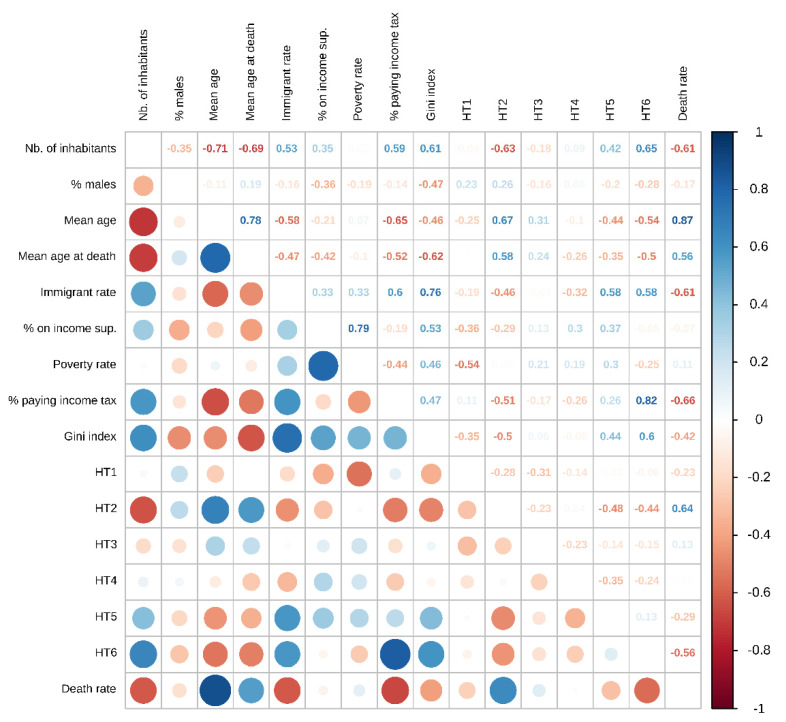
Correlation matrix.

**Figure 3 jpm-11-00627-f003:**
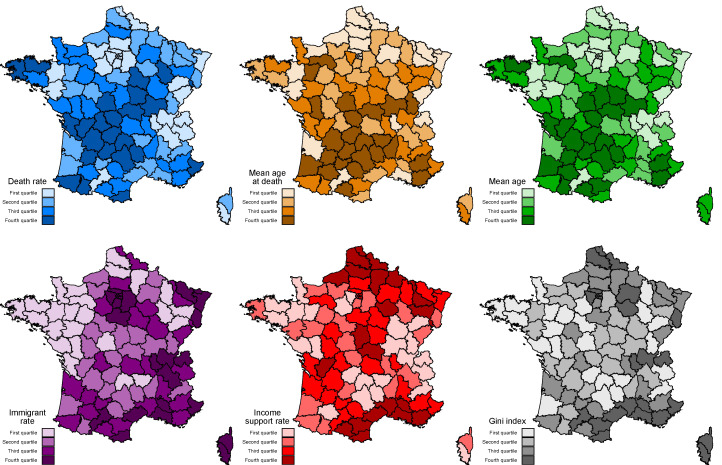
The covariates significantly associated with the two main study outcomes (multivariate analysis). The color intensity represents the four different quartiles of the variable in each French department, in the multivariate analysis.

**Table 1 jpm-11-00627-t001:** Characteristics of the French departments. Variables are quoted for the departments as a whole and as a function of the dichotomized death rates and mean age at death per department. Continuous variables are expressed as the mean ± SD, and categorical variables are expressed as a percentage. The types of “health territory” are described in the Methods section.

Variables		Death Rate			Mean Age at Death		
	All Departements (96)	Departments with a Death Rate < 1% (44)	Departments with a Death Rate ≥ 1% (52)	*p*-Value	Departments with Mean Age at Death < 80 yo (45)	Departments with Mean Age at Death ≥ 80 yo (51)	*p*-Value
Inhabitants	673,324.30 ± 515,271.96	941,296.84 ± 607,946.84	446,578.31 ± 261,859.78	0	970,599.96 ±564,689.75	411,022.25 ±272,279.92	0
Living men percentage	48.5 ± 0.49	48.6 ± 0.49	48.4 ± 0.47	0.041	48.5 ± 0.51	48.5 ± 0.47	0.8624
Mean age	42 ± 2.62	41 ± 2.04	44 ± 1.89	0	41 ± 1.85	44 ± 1.90	0
Proportion of immigrants	7.80 ± 4.57	9.93 ± 5.57	6.00 ± 2.37	0	9.37 ± 5.68	6.42 ± 2.68	0.0013
Proportion of family allowance beneficiaries (%)	17.94 ± 1.91	18.20 ± 2.25	17.72 ± 1.55	0.2239	18.93 ± 1.93	17.06 ± 1.41	0
Poverty rate	14.42 ± 3.03	14.23 ± 3.68	14.59 ± 2.36	0.5654	14.47 ± 3.61	14.38 ± 2.44	0.8869
Median income (k €)	20.7 ± 2.04	21.546 ± 2.627	19.985 ± 0.891	0.0001	21.402 ± 2.431	20.0816 ± 1.368	0.0013
Gini index	0.34 ± 0.03	0.35 ± 0.04	0.33 ± 0.02	0.0006	0.35 ± 0.04	0.33 ± 0.02	0.0001
Proportion of farmers	1.24 ± 0.96	0.77 ± 0.65	1.64 ± 1.01	0	0.69 ± 0.49	1.74 ± 1.01	0
Proportion of craftsmen and company managers	3.69 ± 0.75	3.52 ± 0.76	3.83 ± 0.72	0.0471	3.32 ± 0.64	4.01 ± 0.69	0
Proportion of executives	7.32 ± 4.02	9.21 ± 5.18	5.71 ± 1.34	0	9.14 ± 4.81	5.71 ± 2.19	0
Proportion of white-collar workers	16.07 ± 1.16	16.36 ± 1.47	15.83 ± 0.73	0.0266	16.38 ± 1.32	15.81 ± 0.92	0.0147
Proportion of blue-collar workers	12.98 ± 2.55	12.89 ± 3.13	13.06 ± 1.97	0.7401	12.84 ± 2.85	13.11 ± 2.29	0.5969
Unemployment rate	15.40 ± 2.49	16.59 ± 2.65	14.40 ± 1.83	0	17.06 ± 2.22	13.95 ± 1.66	0
Peri-urban areas * (%)	21.57 ± 16.98	24.62 ± 18.68	19.00 ± 15.10	0.1064	23.03 ± 16.50	20.29 ± 17.45	0.4325
Rural, unattractive areas * (%)	24.37 ± 25.03	13.28 ± 19.05	33.76 ± 25.79	0	11.73 ± 14.25	35.53 ± 27.22	0
Retirement & tourism areas ^$^ (%)	10.14 ± 19.88	6.49 ± 17.11	13.22 ± 21.64	0.099	5.29 ± 12.49	14.41 ± 23.95	0.024
Disadvantaged areas with dedicated governmental health programs * (%)	10.24 ± 16.37	10.51 ± 17.88	10.01 ± 15.15	0.8819	13.66 ± 20.25	7.22 ± 11.33	0.0538
Heterogeneous city centers ^$^ (%)	24.27 ± 18.55	28.46 ± 21.20	20.72 ± 15.28	0.041	30.76 ± 20.31	18.54 ± 14.80	0.001
Wealthy cities and peri-urban areas (%)	9.42 ± 16.65	16.64 ± 21.90	3.30 ± 5.38	0.0001	15.54 ± 20.12	4.02 ± 10.36	0.0005
Death rate	1.01 ± 0.20	0.84 ± 0.14	1.15 ± 0.12	0	0.90 ± 0.17	1.10 ± 0.17	0
Mean age at death	80.03 ± 1.69	79.20 ± 1.60	80.73 ± 1.44	0	78.61 ± 1.22	81.28 ± 0.86	0

*: poor access to healthcare services; ^$^: easy access to healthcare services.

**Table 2 jpm-11-00627-t002:** The death rate, as a function of selected variables. Univariate and multivariate analyses were performed with a tobit model. The types of “health territory” are described in the Methods section. Although the intercepts were estimated for each model, they are not presented in the table for the sake of clarity.

Variables	Univariate Analysis				Multivariate Analysis			
	Coefficient	95% CI		*p*-Value	Coefficient	95% CI		*p*-Value
Inhabitants	−2.36 × 10^−7^	−2.97 × 10^−7^	−1.75 × 10^−7^	0.000	−4.06 × 10^−8^	-1.01 × 10^−7^	2.00 × 10^−8^	0.186
Percentage males	-0.069	−0.149	0.011	0.093	−0.035	−0.086	0.017	0.184
Mean age	0.066	0.059	0.074	0.000	0.054	0.037	0.070	0.000
Immigrant rate	−0.027	−0.034	−0.020	0.000	−0.013	−0.022	−0.003	0.009
Proportion of family allowance beneficiaries (%)	−0.017	−0.066	0.033	0.509	0.048	0.006	0.089	0.025
Poverty rate	0.007	−0.006	0.020	0.271	−0.012	−0.032	0.008	0.232
Percentage of households liable for income tax	−0.021	−0.026	−0.016	0.000	−0.002	−0.011	0.007	0.639
Gini Index	−2.669	−3.828	−1.510	0.000	0.860	−0.384	2.103	0.173
Peri-urban areas * (%)	−0.003	−0.005	0.000	0.018	-	-	-	-
Rural, unattractive areas * (%)	0.005	0.004	0.006	0.000	0.002	0.000	0.003	0.032
Retirement and tourism areas ^$^ (%)	0.001	−0.001	0.003	0.193	0.000	−0.002	0.001	0.758
Disadvantaged areas with dedicated governmental health programs * (%)	0.000	−0.002	0.003	0.860	0.001	−0.001	0.003	0.370
Heterogeneous city centers ^$^ (%)	−0.003	−0.005	−0.001	0.003	0.003	0.001	0.005	0.003
Wealthy cities and peri-urban areas ^$^ (%)	−0.007	−0.009	−0.005	0.000	0.000	−0.002	0.003	0.695

*: poor access to healthcare services; ^$^: easy access to healthcare services.

**Table 3 jpm-11-00627-t003:** Mean age at death, as a function of selected variables. Univariate and multivariate analysis were performed with linear regression. The types of “health territory” are described in the Methods section. Although the intercepts were estimated for each model, they are not presented in the table for the sake of clarity.

Variable	Univariate Analysis				Multivariate Analysis (adj. R^2^ = 0.7594)			
	Coefficient	95% CI		*p*-Value	Coefficient	95% CI		*p*-Value
Inhabitants	−2.27 × 10^−6^	−2.75 × 10^−6^	−1.79 × 10^−6^	0.000	2.32 × 10^−7^	−4.54 × 10^−7^	9.18 × 10^−7^	0.503
Percentage males	0.656	−0.032	1.343	0.062	0.235	−0.349	0.818	0.426
Mean age	0.504	0.423	0.586	0.000	0.444	0.256	0.633	0.000
Immigrant rate	−0.173	−0.239	−0.106	0.000	0.106	−0.002	0.213	0.053
Proportion of family allowance beneficiaries (%)	−0.882	−1.271	−0.493	0.000	−0.520	-0.992	−0.048	0.031
Poverty rate	−0.058	−0.171	0.054	0.311	0.102	−0.122	0.326	0.369
Percentage of households liable for income tax	−0.140	−0.187	−0.094	0.000	−0.030	−0.127	0.067	0.543
Gini index	−33.765	−42.391	−25.138	0.000	−25.437	−39.519	−11.354	0.001
Peri-urban areas * (%)	0.000	−0.020	0.020	0.978	-	-	-	-
Rural, unattractive areas * (%)	0.039	0.028	0.050	0.000	−0.009	−0.026	0.009	0.326
Retirement and tourism areas ^$^ (%)	0.021	0.004	0.038	0.014	−0.001	−0.019	0.017	0.921
Disadvantaged areas with dedicated governmental health programs * (%)	−0.027	−0.047	−0.007	0.008	−0.016	−0.037	0.006	0.155
Heterogeneous city centers ^$^ (%)	−0.032	−0.050	−0.015	0.000	−0.007	−0.028	0.015	0.532
Wealthy cities and peri-urban areas ^$^ (%)	−0.051	−0.069	−0.033	0.000	−0.001	−0.028	0.027	0.955

*: poor access to healthcare services; ^$^: easy access to healthcare services.

## Data Availability

The authors declare that the data collected was gathered from publicly accessible sources: INSEE. Population en 2017 (https://www.insee.fr/fr/statistiques/4515539?sommaire=4516122, accessed on 5 November 2020) [22]; INSEE. Principaux indicateurs sur les revenus et la pauvreté aux niveaux national et local—Principaux résultats sur les revenus et la pauvreté des ménages en 2017 (https://www.insee.fr/fr/statistiques/4507225?som-maire=4507229#consulter-sommaire, accessed on 5 November 2020) [23]; Cafdata. Foyers allocataires percevant le revenu de solidarité active (RSA)—par Commune (http://data.caf.fr/dataset/foyers-allocataires-percevant-le-revenu-de-solidarite-active-rsa-par-commune, accessed on 5 November 2020) [24]; INSEE. Fichier des personnes décédées depuis 1970 (https://www.insee.fr/fr/information/4190491, accessed on 5 November 2020) [25]; Chevillard G, Mousquès J. Accessibilité aux soins et attractivité territoriale: proposition d’une typologie des territoires de vie français [21].

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
