# Peer review of "The Relationship between the Immigrant Rate and Health Status in the General Population in France"

_jpm, 2021, doi:10.3390/jpm11070627_

Round 1

Reviewer 1 Report

Abstract:

  • Line 25: Incomplete sentence (a major issue for what?)
  • Lines 25-26: The description for background doesn't match the objective of the study (factors for immigrants' health status vs. the county-level death rate and mean age at death as a function of the proportion of immigrants). Is it the assumption that the density of immigrants in a county affect immigrants' health status? Was health status measured in terms of (1) death rates and (2) mean age at death?
  • Methods:
    • IV and DV are missing;
    • Why the number of inhabitants is used? It should be a ratio between immigrants and non-immigrants per county.
    • Analysis methods are missing
  • Findings:
    • Line 36: The term ‘immigrant rate’ came out of nowhere. Is it IV, DV, or CV?
    • Line 37: It says “univariate analyses,” but reported “significantly associated,” suggesting the use of analysis method isn’t correct.
    • Line 39: No description of analysis method for the multivariate analysis.
    • Lines 40-41: Needed to describe the direction of association between the death rate and immigrant rate.
  • Interpretation:
    • Lines 44-45: This statement came out of nowhere. That is, there was no statement for socio-economic and health variables in multivariate analysis.
    • Lines 45-26: Counterintuitive results/interpretations since it says that the county with a high proportion of immigrants tended to have lower death rates and the county’s condition for health is poor (i.e., the county with poor health condition tended to have a high proportion of immigrants). However, it interpreted the result as “because immigrants are attracted to economically thriving areas.” This is confusing because it implied economically thriving areas as the areas with poor health conditions.

Introduction

  • In sufficient introduction to current knowledge on immigrant health.

Methods

  • No statement of IV
  • No citations for the sour of the data
  • Lines 103-107: Unclear. For example, if the Chevillard et al.’s classification wasn’t used, how could the authors tell the characteristics of each county?
  • No operationalized form of variables (e.g., in lines 108-109: how the municipality scale was aggregated)
  • Line 111: What do you mean by “Outcomes”? Outcome variables?
  • Line 116: typo à “Student’s t-test”
  • Line 118: “Factors associated with the death rate and the mean age at death” à how can associations be tested in univariate analyses?
  • Line 117: What’s the meaning of ‘the death rates were dichotomized against cut-offs of 1%’? What is the operationalized form? What is the rationale?
  • Lines 117-118: What’s the meaning of ‘the mean age at death were dichotomized against 80 years? What’s the operationalized form? What is the rationale?
  • Line 119: The authors said the mean age at death was dichotomized. Then, what’s the name of linear models for the mean age at death?
  • Lines 119-120: The authors said the death rates were dichotomized. Then, why did they now say “a bounded countinous variable” and use tobit model?
  • Line 123: Is the ‘Access to healthcare facilities’ DV? Then, why didn’t mention it earlier? How is it operationalized?
  • Line 124: The ‘a set of several dependent variables” came out of nowhere. Initially, DVs were (1) Death rates and (2) Mean age at death. Now, they say something else as DVs.

No further comments since the methods section has many places to be improved.

Author Response

Abstract:

According to the reviewer's sound advices, abstract was modified as followed

The abstract could effectively be more representative of the paper. Thanks to the reviewer’s comments, we made significant modifications and the abstract was modified as followed :

Mostly studied at the individual level, the analysis of immigrants’ health status at a populational level may provide a different perspective. Our objective was to investigate at a county-level the relationship between immigrants’ proportion and health status in France, adjusted on health and social determinants.

We analyzed freely accessible databases curated by French public bodies. The dependent variables were death rate and mean age at death. Immigrant rate and other covariates associated with either of the outcomes were explored in univariate and multivariate models. Linear models were used to explain the mean age at death, whereas tobit models were used to explain the death rate.

The immigrant rate varied markedly from one département to another, as did healthcare accessibility, population’s age profile, and economic covariates. Considering univariate models, almost all the studied covariates were significantly associated with the outcomes. The immigrant rate was associated with a lower death rate and a lower age at death. In multivariate models, the immigrant rate was no longer associated with age at death but was still negatively associated with the death rate.

In France, the départements with a higher proportion of immigrants were those with a lower death rate, possibly because immigrants are attracted to economically thriving areas.

Introduction

  • Insufficient introduction to current knowledge on immigrant health.

We thank the reviewer for his comment, significant changes have been made to address this issue.

Methods

  • No statement of IV :

Line 682 : The following statement was added “we considered the death rate and the mean age at death for the dependent variables. As independent variables, we considered the …”.

  • No citations for the sour of the data :

L 676-678  and in the “Data availability statement”: the references to the URL links of the sources used for the article have been added.

  • Lines 103-107: Unclear. For example, if the Chevillard et al.’s classification wasn’t used, how could the authors tell the characteristics of each county?

We agree with the reviewer that without the Chevillard et al.’s classification, characterizing the health access would have been more complex. Other tools (probably less accurate) could have been used, such as the number of general practitioners per inhabitant, the number of hospital.However, we are not sure to understand the point that the reviewer wants to highlight? Even if the Chevillard et al.’s classification was the only tool available, how could such a consideration be viewed as an issue? We cannot say which other tool we would have used, but the fact is that such a Chevillard et al.’s tool was available and chosen to be used.

  • No operationalized form of variables (e.g., in lines 108-109: how the municipality scale was aggregated)

The Chevillard et al.’s  health territories classification is built on the municipality scale but as we considered the département scale, the data was aggregated to obtain for each département its health territories’ composition  expressed as a percentage. The calculations have been weighted with the municipalities characteristics (number of inhabitants).

  • Line 111: What do you mean by “Outcomes”? Outcome variables?

Line 699: We corrected the expression as we indeed meant “Outcome variables”.

  • Line 116: typo à “Student’s t-test”

Line 704: We corrected the typo accordingly.

  • Line 118: “Factors associated with the death rate and the mean age at death” à how can associations be tested in univariate analyses?

We agree that the association between two covariates requires at least two covariates. However, when analyzing such an association using regression models, we consider a dependent covariate (DV), and a unique independent covariate (the so-called IV). We used the expression “univariate analyses” to refer to univariate models (meaning to study the association between two covariates), whereas the expression “multivariate analyses” to refer to multivariate models, meaning models considering one DV and multiple IV). We changed the expression “univariate analysis” in the manuscript by “univariate models” for a better understanding.

  • Line 117: What’s the meaning of ‘the death rates were dichotomized against cut-offs of 1%’? What is the operationalized form? What is the rationale?
  • Lines 117-118: What’s the meaning of ‘the mean age at death were dichotomized against 80 years? What’s the operationalized form? What is the rationale?
  • Line 119: The authors said the mean age at death was dichotomized. Then, what’s the name of linear models for the mean age at death?
  • Lines 119-120: The authors said the death rates were dichotomized. Then, why did they now say “a bounded countinous variable” and use tobit model?

Line 709:

The sentence “the death rate and the mean age at death were dichotomized against cut-offs of 1% and 80 years, respectively” meant that the death rate was dichotomized using 1% as cut-off, to identify départements with a death rate under 1% (which could be expressed as départements with a low death rate) , and those with a death rate above 1% rate (départements with a  high death rate). The mean age at death was dichotomized condering 80 years old as a cut-off, to identify the départements where people died younger ( < 80 y) and those where people died older (> 80y). The death rate (or mean age at death) was calculated for each départements and then classified as low/high depending on the 1% cut-off ( or young age/old age at death depending on the 80 years cut-off).

These transformations were used only for descriptive purposes (in table 1), as we found it interesting to describe the population and the départements as “low/high death rate” and “young/old age at death”. For all the other analyses (and especially for all the univariate and multivariate models), these covariates (death rate and mean age at death) were used as continuous covariates, using linear and tobit models.

We chose to remove this sentence from the article, as it will confuse the readers on the nature of the variables, but we find it useful to keep the dichotomized description in Table 1.

  • Line 123: Is the ‘Access to healthcare facilities’ DV? Then, why didn’t mention it earlier? How is it operationalized?
  • Line 124: The ‘a set of several dependent variables” came out of nowhere. Initially, DVs were (1) Death rates and (2) Mean age at death. Now, they say something else as DVs.

Line 709 :

These comments are extremely right and we apologize for the confusion. In an early version of this manuscript, we also tried to explore the factors associated with “Access to healthcare facilities” using multivariate models. These analyses being quite complexe, we decided to replace them by the study of correlations between each covariates (as presented by the correlation matrix, Figure 2). This paragraph was the remnant of this analysis, and was removed from the manuscript.

Reviewer 2 Report

This paper provides an interesting angle on the link between immigrant populations associated with overall population health. 
I feel the introduction could make it clearer if it is assumed that (after controlling for confounding factors) the differences in population health across regions is a direct result of differences in health of the immigrant populations in these regions?

Author Response

This paper provides an interesting angle on the link between immigrant populations associated with overall population health.

I feel the introduction could make it clearer if it is assumed that (after controlling for confounding factors) the differences in population health across regions is a direct result of differences in health of the immigrant populations in these regions?

We thank the review for his comment. Indeed, as suggested, we modified the introduction to clarify what was already known on the subject and explain the approach we chose to pursue.

Reviewer 3 Report

This paper presents an interesting approach to the use of secondary data for extracting pertinent information in the journal area.

In my opinion, the paper should be accepted with minor revisions namely:

1- In the introduction, it seems important to demonstrate a knowledge of the literature in the area, namely by referencing similar research in France or other countries.

2- The paper presents numerous errors and inconsistencies in formatting and editing. Some examples:

Line 57 - "heterogeneous (3,9)but"

Line 96 - "income support(19),"

Line 98 - "subject to income tax(20)"

Line 100 - "nationality other than French(12)"

Line 64 - "last 70 years (12) : the proportion"

Line 66 - "as a result of the migrant crisis in the 2010s (13) ."

Among many other examples. It should be noted that the use of spaces is currently arbitrary. It should be systematized.

Author Response

This paper presents an interesting approach to the use of secondary data for extracting pertinent information in the journal area.

In my opinion, the paper should be accepted with minor revisions namely:

1- In the introduction, it seems important to demonstrate a knowledge of the literature in the area, namely by referencing similar research in France or other countries.

2- The paper presents numerous errors and inconsistencies in formatting and editing. Some examples:

Line 57 - "heterogeneous (3,9)but"

Line 96 - "income support(19),"

Line 98 - "subject to income tax(20)"

Line 100 - "nationality other than French(12)"

Line 64 - "last 70 years (12) : the proportion"

Line 66 - "as a result of the migrant crisis in the 2010s (13) ."

Among many other examples. It should be noted that the use of spaces is currently arbitrary. It should be systematized.

In line with the reviewer' comment, and with all our apologies, we corrected the typos and formatting incongruences. We also modified the introduction so that the demonstration of our goal was clearer, in particular in line with this issue in France.

Reviewer 4 Report

For the authors’ guidance my evaluation and some constructive remarks that would help to improve the paper’s quality are included below:

---Sophistication of the Argument---

=> The topic area is problematised, the discussion has an obvious structure, moving from a general to a more focused theme(s), ideas are clearly/fully developed, and circular reasoning is not used.

---Appropriate Methodology---

=> There is a clear justification of why the methodology is chosen over alternative approaches. The authors precisely defined objectives, aims, and valid conclusions (e.g., claims are supported with evidence/references made to the literature).

---Clear and Coherent Text---

=> The main ideas presented by the authors are obvious/intelligible and presented in a logical, easy-to-follow manner; the main themes are repeated/summarised; ideas are not “out-of-the-blue” i.e., they develop as a result of the discussion.

---Appropriate Reference to Relevant Published Literature---

=> The major theoretical or empirical work in the field is not omitted; the title of the paper reflects the study/central theme(s); minor to no grammar/spelling mistakes; and appropriate use of scientific/academic voice regarding the relevant research fields. No material included in the submission is copyrighted (e.g., scanned material from books or copied off the internet) or plagiarised; all figures and tables (and any appendices) have suitable titles and their sources are appropriately cited.

---Some Technical Issues & Stylistic Notes---

=> The authors should strictly adhere to the submission guidelines of the Journal of Personalized Medicine.
=> Referencing needs to be improved and/or refined carefully. I recommend the author using grammarly.com and writeandimprove.com to refine the text from minor errors.
=> Avoid one or two-sentence paragraphs.

---Concluding Remarks---

=> Why did the authors not specify specific research inquiries that are relevant to the study? Some follow-up research questions can also be included in the text. In case when research questions are identified, then the authors can attract the attention of readers who are able to conceive the framework of investigation at first glance. During the revision, I recommend the authors to come up with their research questions, and why the research is interesting and relevant to the field. 

=> I recommend the authors clarify their methodology in detail, making sure that their planned methods/research tools are fully detailed. They ought to give attention to justifying their chosen methodology in terms of demonstrating applicability, adjustment, and usefulness in the paper. Thus, I would encourage the authors to undertake some revisions that may take some time. 

=> Consequently, I appreciate there has been a lot of reading and ground covered. However, the study should have a stronger focus, compelling argument and discussion, and an indication of why the paper holds value to the readership of the Journal of Personalised Medicine.

=> Journal of Personalised Medicine is a leading peer-reviewed academic journal. A referee ought to recommend research that is deemed a great contribution to the journal’s future achievements. Consequently, the manuscript - in its current form – demonstrates rigorous research outcomes/findings that can be useful for the readers of the journal.

Author Response

We thank the reviewer for his wised advices and constructive remarks.

---Some Technical Issues & Stylistic Notes---

=> The authors should strictly adhere to the submission guidelines of the Journal of Personalized Medicine.
=> Referencing needs to be improved and/or refined carefully. I recommend the author using grammarly.com and writeandimprove.com to refine the text from minor errors.
=> Avoid one or two-sentence paragraphs.

As recommend, we corrected the manuscript according to the submission guideline, in particular by adapting our referencing. We corrected it and applied the American Chemical Society style as recommended. We also went through the manuscript to correct minor errors. Our manuscript has also been submitted to an English professional translator. Finally, we limited the one sentence paragraphs in particular in the method section.

=> Why did the authors not specify specific research inquiries that are relevant to the study? Some follow-up research questions can also be included in the text. In case when research questions are identified, then the authors can attract the attention of readers who are able to conceive the framework of investigation at first glance. During the revision, I recommend the authors to come up with their research questions, and why the research is interesting and relevant to the field. 

=> Consequently, I appreciate there has been a lot of reading and ground covered. However, the study should have a stronger focus, compelling argument and discussion, and an indication of why the paper holds value to the readership of the Journal of Personalised Medicine.

As required by the reviewers, we try to provide either follow-up question but also to go further in the analysis we can have toward such results. In particular, we tried to enlarge the subject by mentioning the impact that our study can have on policy makers.

In particular, we added two part in the discussion part page 11:

“As the French areas with the highest immigrant rates are those with lowest death rates, French policies should be adapted: specific policies may be needed toward immigrant population in territory that are not the one usually targeted by common programs.”

And

“Managing to include the immigrant time of stay on the host country may help to understand the relationship found. It may also be a useful indicator to evaluate the policies implemented by France to integrate immigrant population either in their health system but also along all the sociodemographic conditions explored.”

We also modified the conclusion part to include this reflexion, as followed :

Even though the relationship between immigration and health status is well known at the individual level, it warrants further investigation at the population level. The territorial distribution of immigrants is closely related to socio-economic and health characteristics. Despite the precarious health conditions of many immigrants, the French areas with the highest immigrant rates are those with lowest death rates. Such results may be useful to adapt health programs toward immigrant population across French territory.

=> I recommend the authors clarify their methodology in detail, making sure that their planned methods/research tools are fully detailed. They ought to give attention to justifying their chosen methodology in terms of demonstrating applicability, adjustment, and usefulness in the paper. Thus, I would encourage the authors to undertake some revisions that may take some time. 

As recommended by the reviewer, we tried to provide the reader with more information either on the tools used than on the methodology applied.

As so, we modified the first part of the method part as followed :

“Data were obtained from freely accessible databases curated by the French National Institute for Statistical and Economic Studies (INSEE 25,26, corresponding to the French population census indicating exhaustively and at the municipality scale the socio-demographic characteristics of the French  population, e.g. number of inhabitants, age, s ex ratio, economical status, percentage of population being born abroad with a nationality other than French), the French Institute for Research and Information in Health Economics (IRDES 24, characterizing at the life-territory level – a supra-municipal scale -the type of access to healthcare for the population) and the French National Family Allowances Office (CNAF 27, reporting  exhaustively and at the municipality scale the part of the population receiving income support).”

We also clarified the statistical analysis part adding information’s about the selection of covariates and the collapse of different scaled data. The part is now written this way :

“The different databases were merged and collapsed at the “department” level by weighting each information collected at the municipal level by the number of inhabitants of this city.

Quantitative variables were quoted as the mean (standard deviation) and were compared using t-tests. Factors associated with the death rate and the mean age at death were explored in univariate and multivariate models. Linear models were used to explain the mean age at death, whereas tobit models were used to explain the death rate (as a bounded continuous variable that cannot be lower than 0% or higher than 100%). No variable selection process was used when performing multivariate models to avoid any overfitting issue. All the considered independent covariables were included in these models.”